# Cellular Toxicity Mechanisms and the Role of Autophagy in Pt(IV) Prodrug-Loaded Ultrasmall Iron Oxide Nanoparticles Used for Enhanced Drug Delivery

**DOI:** 10.3390/pharmaceutics13101730

**Published:** 2021-10-19

**Authors:** L. Gutiérrez-Romero, L. Rivas-García, C. Sánchez-González, J. Llopis, E. Blanco, M. Montes-Bayón

**Affiliations:** 1Department of Physical and Analytical Chemistry, Faculty of Chemistry, University of Oviedo, C/Julián Clavería 8, 33006 Oviedo, Spain; luciaromero@uniovi.es (L.G.-R.); eblancog@uniovi.es (E.B.); 2Instituto de Investigación Sanitaria del Principado de Asturias, (ISPA) Avda. Hospital Universitario S/N, 33011 Oviedo, Spain; 3Biomedical Research Center, Department of Physiology, Faculty of Pharmacy, University of Granada, Avda. del Conocimiento S/N, 18100 Granada, Spain; lorenrivas@ugr.es (L.R.-G.); crissg@ugr.es (C.S.-G.); jllopis@ugr.es (J.L.)

**Keywords:** nanoparticles, cisplatin prodrug, DNA, apoptosis, autophagy, mitochondria

## Abstract

Ultrasmall iron oxide nanoparticles (<10 nm) were loaded with cis-diamminetetrachloroplatinum (IV), a cisplatin (II) prodrug, and used as an efficient nanodelivery system in cell models. To gain further insight into their behavior in ovarian cancer cells, the level of cellular incorporation as well as the platination of mitochondrial and nuclear DNA were measured using inductively coupled plasma mass spectrometry (ICP-MS) strategies. Quantitative Pt results revealed that after 24 h exposure to 20 µM Pt in the form of the Pt(IV)-loaded nanoparticles, approximately 10% of the incorporated Pt was associated with nuclear DNA. This concentration increased up to 60% when cells were left to stand in drug-free media for 3 h. These results indicated that the intracellular reducing conditions permitted the slow release of cisplatin (II) from the cisplatin (IV)-loaded nanoparticles. Similar results were obtained for the platination of mitochondrial DNA, which reached levels up to 17,400 ± 75 ng Pt/ mg DNA when cells were left in drug-free media for 3 h, proving that this organelle was also a target for the action of the released cisplatin (II). The time-dependent formation of Pt-DNA adducts could be correlated with the time-dependent decrease in cell viability. Such a decrease in cell viability was correlated with the induction of apoptosis as the main route of cell death. The formation of autophagosomes, although observed upon exposure in treated cells, does not seem to have played an important role as a means for cells to overcome nanoparticles’ toxicity. Thus, the designed nanosystem demonstrated high cellular penetration and the “in situ” production of the intracellularly active cisplatin (II), which is able to induce cell death, in a sustained manner.

## 1. Introduction

The use of cisplatin as an effective anticancer drug for treating various malignancies has been well-established over the years [1]. Although its antineoplastic effects were initially ascribed to its ability to generate unrepairable nuclear DNA lesions, increasing evidence also associates its mode of action with the alteration of both nuclear and cytoplasmic signaling pathways [2]. However, the positive chemotherapeutic effect of cisplatin in most cancers (e.g., ovarian, prostate, etc.) is hampered by inherent and acquired drug resistance, a multifactorial and still not well characterized process [3]. In fact, several mechanisms have been suggested to participate in conferring platinum-resistant properties to a tumor cell, such as genetic alterations in genes involved in drug uptake (and efflux), DNA repair, autophagy [4], apoptosis, and cell cycle control pathways [5].

Regarding drug uptake, the entrance of cisplatin inside cells is predominantly driven by the copper transport receptor 1 (CTR1) [6]. One way to circumvent incorporation through receptors like CTR1 is through the use of endocytic routes [7]. For this purpose, combinations of cisplatin (or cisplatin precursors/prodrugs) to nanostructures that permit efficient drug uptake are being widely investigated [8,9,10]. Among them, nanodelivery systems including liposomes or metallic nanoparticles show the capability to incorporate cisplatin (IV) prodrugs [9,11,12,13]. Upon entrance into the cell cytosol, the reduction of the platinum (IV) center to platinum (II) by glutathione or ascorbate occurs, resulting in the formation of functional cisplatin (II) [14,15]. This reaction occurs more efficiently in the reducing environment of cancer cells, increasing the selectivity of the treatment [16,17].

Beside reduced intracellular incorporation, cellular resistance to cisplatin can also result from other mechanisms, including enhanced nuclear DNA repair by the nucleotide excision repair (NER) system or even evasion of cell death by apoptosis [3] or autophagy [18]. Autophagy plays a key role in the quality control of cellular components and can be induced by a number of stress conditions, including drug treatment. Recent studies have indicated that acute cisplatin treatment activates an autophagic response that serves as a survival factor to counteract cisplatin-induced cell death, increasing cell drug resistance [19]. To minimize all these routes of resistance, various alternatives have been proposed regarding drug design. On the one side, the use of Pt drugs that attack mitochondrial DNA (mtDNA), rather than nuclear DNA where NER machinery is inactive, represents an attractive alternative to induce cell apoptosis [20,21]. The use of (nano-)delivery systems that permit controlled drug release and thus eliminate the problems associated with conventional therapy can also be suggested.

Previously obtained results in our group revealed that ultrasmall iron oxide nanoparticles (<10 nm) coated with tartaric and adipic acids could be a good vehicle to transport cisplatin (IV) prodrugs. When cells were exposed to these nanoparticles (without the prodrug), they could be easily taken up (up to 4 mM Fe concentration levels) without signs of cell toxicity [21]. In addition, it was observed that their subcellular fate was mainly related to mitochondria, affecting their respiratory and glycolytic parameters, inducing the migration of the cellular state towards quiescence, and promoting and triggering the autophagic process [4]. Therefore, the loading of these nanoparticles with Pt(IV)- prodrugs was expected to facilitate drug penetration into cells by using endocytic routes and the slow release of the drug locally inside the intracellular targets (nDNA and mtDNA). These effects altogether were expected to induce a more efficient apoptotic pathway when using the Pt(IV)-loaded nanoparticles than when using cisplatin directly.

Therefore, the present work shows the advantages of using ultrasmall iron oxide nanoparticles (<10 nm) loaded with cisplatin (IV) prodrug compared to using cisplatin. Evaluation of the drug uptake in different cell lines of ovarian cancer, the access to molecular targets (nDNA and mtDNA) and toxicity pathways in terms of cell viability, the level of apoptotic and autophagic cells, and cell cycle arrest are studied.

## 2. Materials and Methods

### 2.1. Synthesis of Pt(IV)-Loaded Iron Oxide Nanoparticles

Iron (III) chloride hexahydrate (98%, Sigma-Aldrich, Madrid, Spain) was used as a nanoparticle precursor. Sodium tartrate dihydrate (99–101%, Sigma-Aldrich) and adipic acid (99%, Sigma-Aldrich) were solubilized in 0.9% potassium chloride (Merck, Darmstadt, Germany) solution and used as coating agents. Ammonium acetate (>98%, Sigma-Aldrich) was used for the synthesis buffer and 5 mol·L^−1^ sodium hydroxide (Merck) was prepared for nanoparticle precipitation. All working standard solutions were prepared using 18 MΩ·cm de-ionized water obtained from a PURELAB Flex 3 system (ELGA Veolia, Lane End, UK). Cis-diamminetetrachloroplatinum (IV) (99.9%) was obtained from Sigma-Aldrich. The synthesis and characterization of the Pt(IV)-loaded nanoparticles (Pt(IV)-FeNPs) followed previously published work [6]. In brief, the iron nanoparticles were first synthesized following previous publications [22] by precipitation of Fe^3+^ in the presence of a highly basic medium (5 mol·L^–1^ NaOH solution) with the addition of tartrate and adipic acid solution. For incorporation of cis-diamminetetrachloroplatinum (IV) (prodrug), a solution of 5 mmol·L^–1^ of the prodrug was incubated with the particles for 6 h at room temperature. The excess of the prodrug was eliminated by ultracentrifugation using a 3000 Da Ultra-15 MWCO centrifugal filter. The quantification of the level of loading of the prodrug onto the particles was conducted with HPLC-ICP-MS by monitoring Pt and Fe simultaneously [23]. Each batch of modified nanoparticles was used for 2 weeks and then discarded. During this period, they proved to be stable.

### 2.2. Cell Conditions

The A2780 cell line (ovarian carcinoma) was kindly provided by Dr. J.M. Pérez Freije (Department of Biochemistry and Molecular Biology, University of Oviedo, Spain) and the human ovarian carcinoma cell line OVCAR-3 was purchased from the American Type Culture Collection (ATCC, Manassas, VA, USA). Both lines were authenticated at the Biotechnological and Biomedical Assay Unit of the Scientific and Technical Services (SCTs) at the University of Oviedo. The A2780 cell line was grown in RPMI 1640 medium (Gibco, Waltham, MA, USA) supplemented with 10% fetal bovine serum (FBS) (Gibco, Waltham, MA, USA) and 5 μg·mL^–1^ Plasmocin Prophylactic (InvivoGen, San Diego, CA, USA). The OVCAR-3 cell line was also grown in RPMI medium but supplemented with 20% FBS, 0.01 mg·mL^–1^ of bovine insulin (Sigma Aldrich), and 5 μg·mL^–1^ Plasmocin Prophylactic. All cells were grown at 37 °C in a 5% CO_2_ atmosphere.

### 2.3. DNA Isolation

Isolation of DNA from A2780 was conducted using the PureLink™ Genomic DNA Mini Kit (Invitrogen, Carlsbad, CA, USA) silica-based column DNA purification kit. The kit was used according to the manufacturer’s instructions with the inclusion of RNAse, a treatment to generate RNA-free genomic DNA. The extracted DNA was eluted using 100 μL of the elution buffer (10 mM Tris-HCl, pH 9.0, 0.1 mM EDTA). The isolation of mtDNA was performed using a Mitochondrial DNA Isolation Kit™ (Abcam, Cambridge, UK). This kit provides a powerful tool for isolating mtDNA with high yield and purity and without contaminations from genomic DNA, and it was used according the manufacturer’s instructions.

DNA purity was confirmed by comparing the ratio of UV measurements at 260 and 280 nm with the ratio of pure DNA standards. The concentration of isolated DNA was determined by ^31^P^16^O^+^ monitoring using ICP-MS after sample digestion using 80 μL of HNO_3_ (65%) and 80 μL of H_2_O_2_ (30%). The sample was further diluted to 1:10 for analysis. A ^31^P^16^O^+^ calibration curve was constructed by applying the same digestion procedure to commercially available calf thymus DNA. Simultaneously, ^195^Pt was measured within the same experiments. The final results are given as ng Pt/mg DNA.

### 2.4. ICP-MS Analysis

All ICP-MS experiments in this study were performed using the triple quadrupole instrument iCAP TQ ICP-MS (Thermo Fisher Scientific, Bremen, Germany) in single quadrupole (SQ) mode for ^195^Pt^+^ monitoring. For phosphorous measurements, the formation of ^31^P^16^O^+^ was achieved by pressurizing the cell with O_2_. For the chromatography experiments (required for the characterization of the particles) as well as for Pt measurement in DNA, the ICP-MS was fitted with a cyclonic spray chamber and a conventional concentric nebulizer.

### 2.5. Cytotoxicity Experiments

The CellTiter-Blue™ Cell Viability Assay from Promega (Madison, WI, USA) was carried out in order to assess the cytotoxicity of FeNPs-Pt(IV) in the A2780 cell line. Cells grown in a 96-well flat bottom plate were incubated with FeNPs-Pt(IV) to obtain different concentrations ranging from 0 to 30 μM over 24 h of treatment. Similar experiments were conducted by maintaining the cells after exposure for 3 more hours in drug-free RPMI 1640 medium. After the incubation, the nanoparticles and the floating cells were removed and clean medium mixed with CellTiter-Blue™ Reagent (resazurin dye, 20 μL) was added to each well. The plate was shacked for 10 s and incubated using standard cell culture conditions for 3 h. The fluorescence was measured at 560/590 nm using the microplate reader. A microplate fluorescence reader, the Varioskan Flash Spectral Scanning Multimode Reader (Thermo Fisher Scientific), was employed for this purpose. Every treatment at every concentration was done in triplicate and the cytotoxicity results are given as the means with the error bars corresponding to the standard deviation. The results for the cytotoxicity induced in A2780 for the nanoparticles without prodrugs are shown in Appendix A.

### 2.6. Transmission Electron Microscopy

After 24 h exposure to the Pt(IV)-loaded nanoparticles, cells were then fixed with glutaraldehyde and formaldehyde at 4 °C for 1 h, followed by dehydration with ascending series of alcohol and embedding of the samples in epoxy resin. All these experiments followed the sample preparation protocol used to address the incorporation of titanium dioxide nanoparticles in previous publications [24].

### 2.7. Cell Death Mechanism

#### 2.7.1. Apoptosis/Necrosis

Cell apoptosis was measured following the Annexin V-FITC Apoptosis Detection Kit (Sigma Aldrich) protocol. For this purpose, cells grown in a 96-well flat-bottom plate were incubated with the labelled nanoparticles to achieve a final concentration of 20 µM of the FeNPs-Pt(IV) over 24 h. Similar experiments were conducted by maintaining the cells after exposure for 3 more hours in drug-free RPMI 1640 medium. Then, 200 µL of a binding buffer (5 × 10^5^ cells), 5 µL of Annexin V FITC conjugate, and 10 µL of propidium iodide (stock of 20 µg·mL^−1^) were added to the cells. They were incubated at room temperature in the dark for 10 min and then they were measured by flow cytometry, counting 1 × 10^4^ events. For these measurements, a CytoFLEX S Flow Cytometer (Beckman Coulter, Brea, CA, USA) was used. The results for the cell necrosis and apoptosis induced in A2780 for the nanoparticles without prodrugs are shown in Appendix A.

#### 2.7.2. Autophagy Induction

Autophagy in A2780 cells was evaluated by flow cytometry using the CYTO-ID ENZKIT175 Autophagy Detection Kit 2.0 (Enzo Life Sciences, Lausen, Switzerland) following the manufacturer’s instructions. In brief, 1 × 10^5^ cells/mL were seeded in 6-well plates and incubated with the conditions described above for 24 h with 20 µM of FeNPs-Pt(IV). Similar experiments were conducted by maintaining the cells after exposure for 3 more hours in drug-free RPMI 1640 medium. A total of 500 nM of rapamycin was used as the positive control. Trypsin was used to detach cells from the wells, and Assay Buffer 1× was used for washing. Cells were incubated in RPMI 1640 medium without phenol red indicator and with green stain solution for 30 min at room temperature. Cells were collected, washed with Assay Buffer 1×, and suspended in Assay Buffer 1× before being analyzed in a flow cytometer (Becton Dickinson Biosciences, East Rutherford, NJ, USA). The software FACS Diva 6.1 (Becton Dickinson Biosciences) was used to analyze the results of the experiments. Furthermore, p62 levels were studied using an ELISA kit (Enzo Life Sciences, Lausen, Switzerland), according to the instructions provided by the manufacturer. The experiment was performed in triplicate.

### 2.8. Cell Cycle Arrest Study

The cell cycle was measured following the Invitrogen protocol. Briefly, cell suspensions (5 × 10^5^ cells previously exposed to 20 µM of FeNPs-Pt(IV) for 24 h or 24 h plus a recovery time in drug-free media for 3 more hours) were fixed with EtOH 70%, washed with PBS, and 100 µL of PBS was added. A solution of RNAse (120 µL of 100 µg·mL^–1^) and propidium iodide (120 µL of 140 µg·mL^–1^) was added and the solution was incubated for 30 min. Cells were measured by cytometry counting 1 × 10^4^ events and the data were analyzed with Modfit Lt (Verity Software House, ME, USA) software. The results for the cell necrosis and apoptosis induced in A2780 for the nanoparticles without prodrugs are shown in Appendix A.

### 2.9. Statistical Analysis

Statistical analysis was performed using Student’s t-test and a *p*-value of <0.05 was considered as significant. In addition, the F-test was performed to compare variances among datasets. In the cases when F_experimental_ < F_critical_ (F_exp_ < F_cri_) and, thus, no significant differences among SDs were detected, Student’s t-test was applied. When F_exp_ > F_cri_, instead of Student´s *t*-test, the Welch test was applied.

## 3. Results and Discussion

### 3.1. Quantitative Platination Studies of Nuclear and Mitochondrial DNA

The use of nanoparticles in medicine could lead to the development of new treatment strategies and tools for therapy. Thus, previous work has demonstrated that iron nanoparticles have the capacity to be efficient carriers of moieties such as drugs or genes due to their biocompatibility, easy synthesis, and reduced size [25]. In this study, the conjugation of cisplatin (IV) prodrugs to ultrasmall FeNPs was expected to provide very efficient incorporation of the drug into the cell cytosol through endocytic routes. Thus, A2780 cells were treated with 20 µM Pt (in the form of Pt(IV)-FeNPs), and the TEM results are plotted in Figure 1. It was possible to observe the presence of the Pt(IV)-FeNPs within the cells either individually or in small aggregates (yellow circles; elemental analysis is provided in the Appendix A), which might have been due to the formation of an early endosome through a mechanism involving an endosomal pathway, as previously described [4,26]. Such mechanisms can be used to improve drug internalization and help to circumvent the intracellular pathways and biological barriers of the free drugs.

To further evaluate the enhanced transport of the Pt(IV) prodrug loaded onto the nanoparticles, quantitative Pt incorporation results were calculated using ICP-MS. The results obtained for A2780 cell line revealed average concentrations of about 12.3 ± 2.1 fg Pt/cell versus 3.17 ± 0.76 fg Pt/cell when using cisplatin directly (see the Appendix A). Considering a Pt concentration of 20 µM in both cases, the levels taken up corresponded to 0.15% and 0.05%, respectively, of the Pt given to the cells. Thus, the use of Pt(IV)-loaded nanoparticles increased the drug incorporation levels by a factor of three to four. Such differences were not as dramatic in other tested ovarian cancer cells (e.g., OVCAR-3) exposed to 20 µM (see the Appendix A), which showed significantly lower sensitivity to cisplatin (IC_50_ 30 µM for cisplatin in this cell model). In the case of the A2780, the enhanced transport efficiency provided by the nanoparticles with respect to free cisplatin would have reduced the drug concentration required to obtain comparable cellular toxicity. In addition, the activation of the pre-target resistance mechanisms (like the increased detoxification and efflux of the drug) might have been decreased due to the slow release of the drug from the nanoparticle surface [27,28].

Once in the cytosol, the pathway taken by the Pt(IV) prodrug to reach nuclear DNA requires its reduction to Pt(II) in the presence of reducing molecules like ascorbic acid or glutathione, with the consequent loss of the axial ligands [14]. This process should produce functional cisplatin (II) that can reach the cell nucleus or the mitochondria to interact with DNA, forming the so-called DNA–cisplatin adducts. Firstly, the isolation of nuclear DNA was conducted, and the platination results can be observed in Figure 2. As can be seen, immediately after exposure (t = 0 h), cells showed about 250 ± 50 ng Pt/mg DNA. Taking into account the intracellular Pt (12.3 ± 2.1 fg Pt/cell) and the estimated amount of DNA per cell (about 6 pg DNA/cell), the maximum expected concentration of the drug associated with DNA would be 2500 ng Pt/mg DNA. Thus, the observed 250 ± 50 ng Pt/mg DNA represented about 10% of the cytosolic Pt bound to DNA. This concentration increased significantly (by about fivefold) when the cells were left to stand for 3 h in drug-free media (t = 3 h), reaching levels of approximately 1300 ng Pt/mg DNA, which represented about 60% of the Pt taken up. These results can be explained by the time-dependent release of cisplatin from the Pt(IV)-FeNPs in the cell cytosol, as previously observed in model solutions [23]. In the case of the OVCAR-3 cell line, despite the fact that the cell uptake was significantly lower, the DNA-bound Pt concentration (28 ng Pt/mg DNA) after 24 h exposure also corresponded to approximately 10% of the concentration taken up (1.6 fg/cell). 

When the cells were left to stand for 12 h and 24 h respectively in drug-free media (t = 12 h and t = 24 h), the Pt-DNA adducts could be repaired thanks to some of the existing mechanisms for cells survival (e.g., NER), thus reducing the Pt detected in DNA. The treatment with cisplatin at the same concentration, on the other hand (see Figure 2), resulted in the highest Pt concentration in DNA (50 ng Pt/ mg DNA) at the end of the exposure period. This corresponded to 10% of the cisplatin incorporated in the cytosol (a similar percentage to what was found when using the Pt(IV)-FeNPs), followed by a decrease after 3 (4%), 12 (7%) and 24 h (3%), possibly due to the repair of the formed adducts.

Similar measurements were taken for mtDNA (after isolation with the specific kit as detailed in the Materials and Methods section) and the results can be seen in Figure 3. As in the case of nDNA, the Pt concentration in mtDNA was significantly higher when using the Pt(IV)-FeNPs (8400 ± 55 ng Pt/ mg DNA) than when using cisplatin directly (108 ± 8 ng Pt/mg DNA).

As previously stated, initial experiments conducted with these nanoparticles revealed that their subcellular fate was related to mitochondria [4]; thus, the high concentration of Pt in this organelle was in some sense expected. Furthermore, after t = 3 h in drug-free media, the concentration was twofold higher (up to 17,400 ± 75 ng Pt/ mg DNA), showing that the platination of mtDNA when using Pt(IV)-FeNPs was also more efficient here than when using cisplatin directly and required, as in the case of nDNA, a time lapse. Therefore, once cisplatin was released from the nanoparticles, the quantity of adducts formed with mtDNA increased by as much as 30-fold compared to nDNA, and this must be considered a significant contribution to cell toxicity, as proposed by other authors [19,29].

### 3.2. Cellular Viability, Apoptosis, and Cell Cycle Arrest

A cellular viability test was conducted to evaluate the effect of the Pt(IV)-loaded nanoparticles at different concentrations in the A2780 cell model immediately after 24 h of exposure (t = 0) and after different durations of cell rest in drug-free media. These results are collected in Figure 4.

As can be seen, there were two different trends in the obtained data for the exposure concentrations: from 0 to 5 µM, a progressive decrease in cell viability occurred for cells measured at t = 0 (blue dots) and, similarly, after t = 3 h in drug-free media (red dots). However, the cell viability remained almost constant (about 100%) when cells were measured after 24 h in drug-free media (t = 24 h, grey dots) in this range of concentrations. By comparing these results with the DNA platination results, it is possible to associate the decrease in cell viability at t = 0 and t = 3 h with the increase of DNA platination (see Figure 2). Apparently, after 3 h in drug-free media, the Pt species were still being released from the nanoparticle surface due to the cytosolic reducing agents, forming the so-called DNA adducts and inducing a decrease in cell viability. However, when cells were left to stand for 24 h in drug-free media (grey dots), the DNA repairing mechanism started to act, counteracting the formation of the adducts with their elimination (see Figure 2). Thus, only higher concentrations of the Pt(IV)-loaded nanoparticles (>5 mM) demonstrated a significant decrease in cell viability. Taken together, these results reveal the importance of the kinetics of the different processes occurring inside the cell in relation to the Pt(IV)-loaded nanoparticles for the elucidation of the toxicity mechanisms. 

To further elucidate whether the platination of DNA (both nDNA and mtDNA) resulted in in the activation of the apoptotic pathway, as occurs when cells are treated with cisplatin [30], cells were tested for apoptosis/necrosis. For this purpose, cells were stained with propidium iodate and Annexin V. Annexin V specifically binds phosphatidylserine in the cell membrane in which the cells entered into the apoptotic phase, whereas PI stains DNA in necrotic cells where the cell membrane disintegrates. The results of cell exposure to 20 µM of the Pt(IV)-loaded nanoparticles at t = 0 and t = 3 h in drug-free media can be seen in Figure 5A,B, respectively. As can be observed, Pt(IV)-loaded particles induced apoptosis/necrosis in A2780 cells in a time-dependent manner. At t = 0 h, cell damage was induced, accounting for approximately 14% apoptosis and about 5.5% necrosis compared to the control cells. At t = 3 h, the percentage of apoptotic cells increased significantly up to about 22%, while necrotic cells remained constant or even decreased slightly. These results were in agreement with the cell viability tests and also with the DNA platination results, confirming that when using the Pt(IV)-loaded nanoparticles, the main mechanism of cell toxicity seemed to be the apoptotic route, as in the case of cisplatin.

The formation of Pt-DNA adducts was found to block DNA replication, which could subsequently cause inhibition of cell proliferation. Therefore, to complete the study on the Pt(IV) loaded nanoparticles, it was necessary to address the question of whether they exerted the same effect on the cell cycle. Cell cycle analysis through quantitation of DNA content takes into account the fact that cells that are in the S phase have more DNA than cells in G1; therefore, they take up proportionally more dye and fluoresce more brightly. The cells in G2 are approximately twice as bright as cells in G1. Figure 6 shows the results corresponding to the cell cycle arrest in control cells, at t = 0, and at t = 3 h of cell rest in drug-free media. As can be seen, upon exposure (t = 0), among the cell cycle phases, the S phase was the most severely affected by the treatment with the Pt(IV)-loaded nanoparticles that occurred at the earliest times. More cells were arrested in the S phase (30%) compared to the control (10%).

These results are consistent with those from previous publications [31] that also demonstrated that slowed progression through the S phase was the most important cytokinetic event in the case of cisplatin treatment. In addition, the same authors showed that the cell cycle perturbations correlated with the level of DNA damage, mainly in the form of DNA adducts. This was also observed in this case when using Pt(IV)-loaded nanoparticles. However, the cell cycle arrest in the S phase after t = 3 h in drug-free media, although statistically significant, did not show a dramatic increase compared to that observed at t = 0 (34% versus 30%). Since the S phase corresponds to the phase in which DNA is synthesized, the number of adducts formed at t = 0 seems to have been high enough to prevent DNA replication, thus inducing cell cycle arrest in this phase and inhibiting the progression to G2 even more.

### 3.3. Autophagy

Autophagy is a highly conserved process by which cytoplasmic components are sequestered in double membrane vesicles called autophagosomes and degraded upon fusion with lysosomal compartments. In fact, recent studies have indicated that acute cisplatin treatment activates an autophagic response that serves as a survival factor to counteract cisplatin-induced cell death. In this case, the Pt(IV)-loaded nanoparticles must be studied as potential inductors of autophagy as a cell defense mechanism or, on the contrary, as “Trojan horses” introducing Pt intracellularly more efficiently than the administration of cisplatin. For this purpose, cells exposed to 20 µM of Pt(IV)-loaded nanoparticles at t = 0 and at t = 3 h after cell rest in drug-free media were measured for the production of autophagosomes. To achieve this, fluorescence signaling was performed using two dyes: Hoechst 33342 dye and Cyto-ID^®^ Green dye. As can be observed in Figure 7A, the treatment of A2780 cells with the Pt(IV)-loaded nanoparticles led to a higher content of autophagic vacuoles compared to the control group (6% versus 1%), but it was significantly lower than that found in cells treated with the inductor of autophagy (rapamycin). Moreover, after t = 3 h in drug-free media, the number of autophagic cells decreased from 6% to 3%.

In addition, to determine whether the autophagic flux was initiated by the Pt(IV)-FeNPs, p62 was determined (see Figure 7B). As can be seen in the figure, p62 decreased when cells were treated with Pt(IV)-FeNPs, producing a similar effect as rapamycin (positive control). Similar to the formation of autophagic vacuoles, when cells were left in drug-free media (t = 3 h) this effect reversed to the level of the control sample. Taken together, these results can be explained as an initial cellular response to eliminate the endocyted particles (at t = 0 h). Afterwards, the activation of the apoptosis routes (at t = 3 h) to induce irreversible cell damage could not be counteracted by the formation of the autophagic vacuoles and, therefore, the initial values decreased and the levels of p62 were reestablished.

By comparing these results with those previously observed when evaluating the “naked” nanoparticles (without loading them with the prodrug) [4], it can be concluded that the induction of autophagy can be ascribed to the presence of the nanoparticles and not to the loading with the prodrug. Thus, these data confirmed that the autophagic process can be modulated by ultrasmall iron nanoparticles themselves, providing an excellent and efficient platform for the development of a new drug delivery nanosystem to overcome autophagy-mediated cisplatin resistance.

## 4. Conclusions

The viability of using ultrasmall iron oxide nanoparticles to enhance intracellular delivery of a Pt(IV) prodrug was proved by observing the presence of the nanoparticles in the cell cytosol using TEM and, quantitatively, by measuring Pt using ICP-MS. The incorporated prodrug undergoes further intracellular reduction to form functional cisplatin (II) and its interaction with DNA (nuclear and mitochondrial) is superior to that of cisplatin. This mechanism, however, was found to be time-dependent, achieving maximum platination levels for both nDNA and mtDNA after the cells were left in drug-free media for 3 h following the exposure period. The level of the formation of adducts could be correlated with the decrease in the cell viability, even at lower concentrations, at t = 0 h and t = 3 h after exposure. Longer times in drug-free media (t = 24 h) started to show the effect of the cellular repair mechanisms (e.g., NER), yielding higher cell viability. The main mechanism of cell damage, once the adducts were formed, was apoptosis. In addition, cell arrest in the S phase confirmed that there was analogous behavior to that of cisplatin. In summary, the use of a nanodelivery system like ultrasmall iron oxide nanoparticles can serve to enhance prodrug penetration in cells, allowing the reduction of the employed doses, which are often the cause of systemic toxicity. The prodrug undergoes time-dependent intracellular reduction to cisplatin (II), and this phenomenon occurs more efficiently in the reducing environment of tumor cells. The main advantage is the “in situ” production of the intracellularly active cisplatin (II), which is able to induce cell death. Collectively, these results suggest that ultrasmall iron oxide nanoparticles, due to their small diameter, can be used to deliver cisplatin into cancer cells in a sustained manner in the long term, avoiding multiple administrations.

## Figures and Tables

**Figure 1 pharmaceutics-13-01730-f001:**
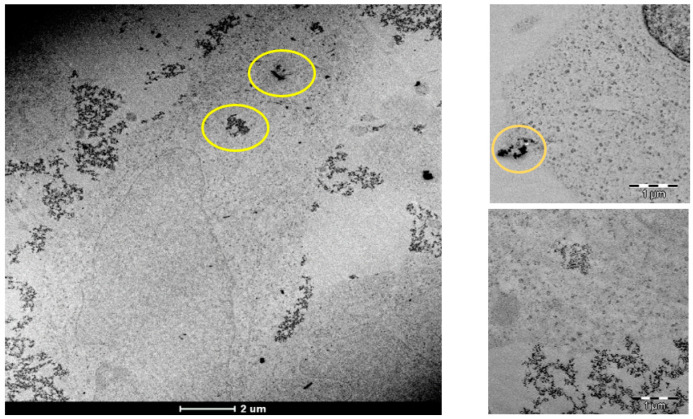
Transmission electron microscopy (TEM) of cells after exposure to 20 µM Pt as Pt(IV)-loaded iron oxide nanoparticles for 24 h (nanoparticles marked with yellow circles).

**Figure 2 pharmaceutics-13-01730-f002:**
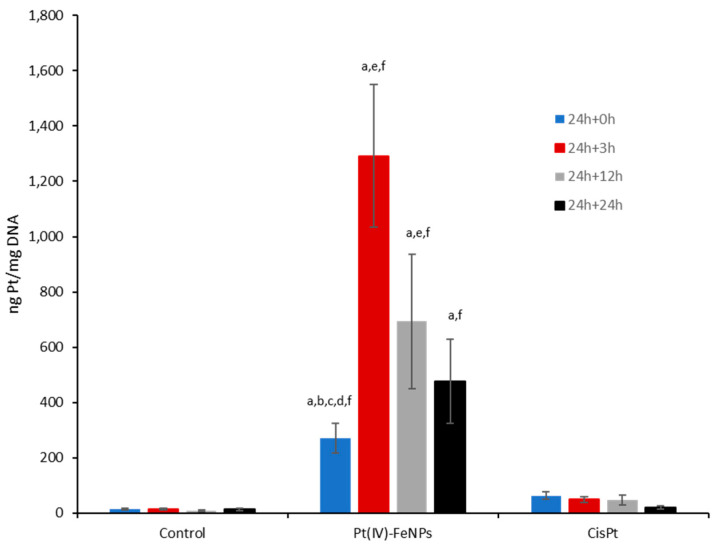
Accumulation of Pt in DNA after exposure to 20 µM Pt as Pt(IV)-loaded iron oxide nanoparticles for 24 h; t = 0 h corresponds to the measurement right after exposure, t = 3 h corresponds to the measurement after 3 h cell rest in drug-free media, t = 12 h and t = 24 h correspond to cell rest for 12 h and 24 h respectively. The same experiments were undertaken with cisplatin. *p* < 0.05 (**a**) against control; (**b**) against Pt(IV)-FeNPs 24 h treatment, t = 0 h; (**c**) against Pt(IV)-FeNPs, t = 3 h; (**d**) against Pt(IV)-FeNPs, t = 12 h; (**e**) against Pt(IV)-FeNPs, t = 24 h; (**f**) against cisplatin 24 h treatment.

**Figure 3 pharmaceutics-13-01730-f003:**
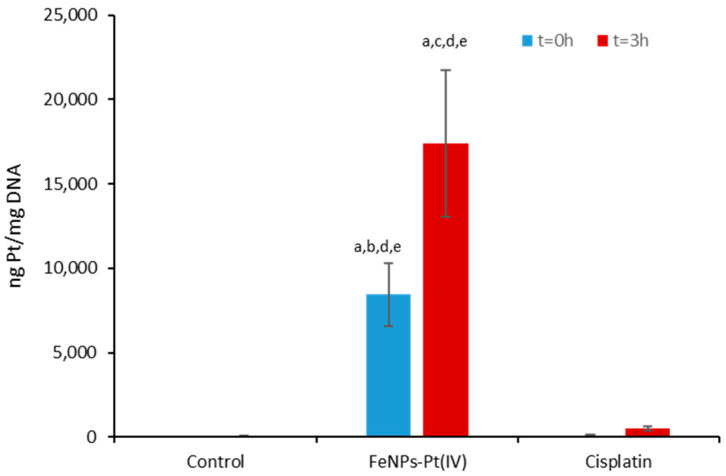
Accumulation of Pt in mitochondrial DNA after exposure to 20 µM of Pt(IV)-loaded iron nanoparticles for 24 h (t = 0 h) and after leaving cells in drug-free media for 3 h (t = 3 h). The same experiments were performed with cisplatin. *p* < 0.05 (**a**) against control; (**b**) against Pt(IV)-FeNPs (t = 0 h); (**c**) against Pt(IV)-FeNPs (t = 3 h); (**d**) against cisplatin (t = 0 h); (**e**) against cisplatin (t = 3 h).

**Figure 4 pharmaceutics-13-01730-f004:**
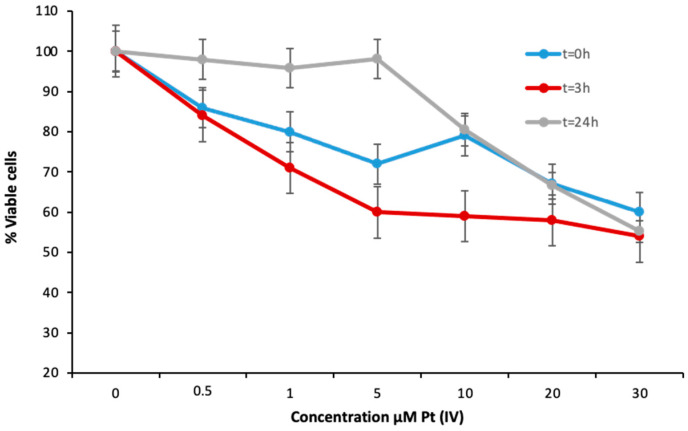
Cell viability results evaluated for A2780 cells treated with Pt(IV)-loaded iron oxide nanoparticles at concentrations ranging from 0 to 30 μM for 24 h (t = 0 h) and after cell rest in drug-free media for 3 h (t = 3 h) and 24 h (t = 24 h).

**Figure 5 pharmaceutics-13-01730-f005:**
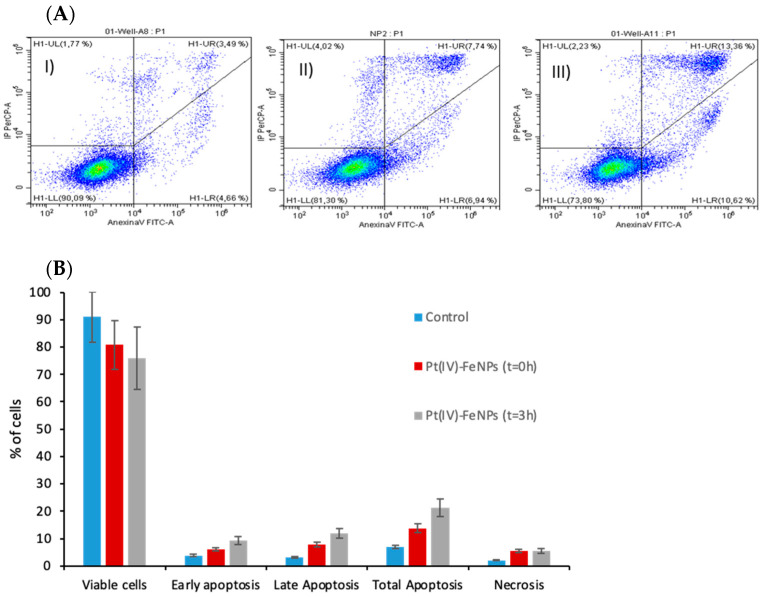
(**A**) Viable, apoptotic, and necrotic cell flow cytometry results for (**I**) control and (**II**) cells exposed to 20 µM of iron nanoparticles covered by cisplatin (IV) prodrug for 24 h and (**III**) t = 3 h in drug-free media. (**B**) Graphic representation of cell percentages at each cell stage. For viable cells and total apoptosis, the results were statistically different against control at t = 0 and t = 3 h for *p* < 0.05.

**Figure 6 pharmaceutics-13-01730-f006:**
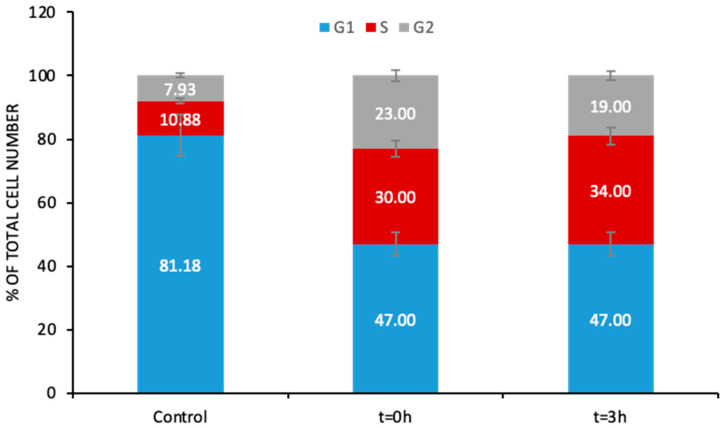
Cell cycle percentages for each cycle stage for cells treated with 20 µM Pt as Pt(IV)-loaded iron oxide nanoparticles for 24 h (t = 0 h) and after cell rest in drug-free media (t = 3 h). For G1, S, and G2, the results were statistically different against control for *p* < 0.05.

**Figure 7 pharmaceutics-13-01730-f007:**
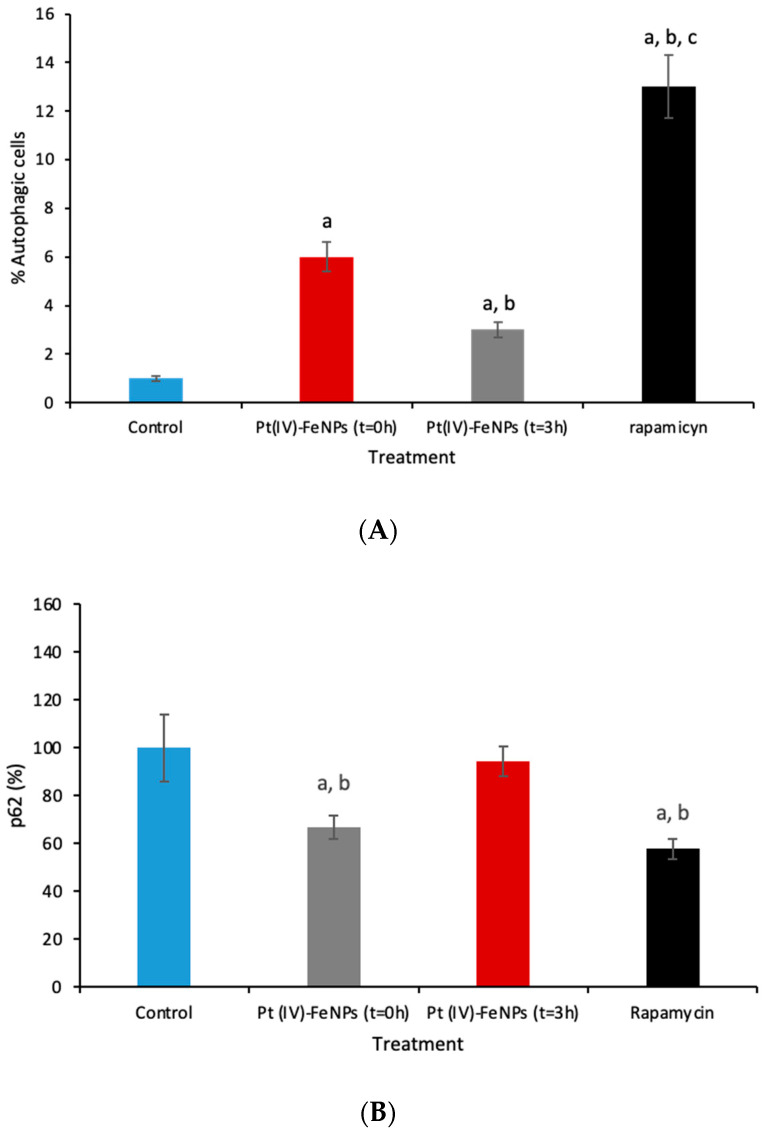
(**A**) Induction of autophagy evaluated by flow cytometry for cells treated with 20 µM Pt as Pt(IV)-loaded iron oxide nanoparticles for 24 h (t = 0 h) and after cell rest in drug-free media (t = 3 h): (**a**) vs. control; (**b**) vs. Pt(IV)-FeNPs (t = 0 h); (**c**) vs. Pt(IV)-FeNPs (t = 3 h). (**B**) Percentage of p62 (compared to control cells) in A2780 cells treated with 20 µM Pt as Pt(IV)-loaded iron oxide nanoparticles for 24 h (t = 0 h) and after cell rest in drug-free media (t = 3 h): (**a**) vs. control; (**b**) vs. Pt(IV)-FeNPs (t = 3 h).

## Data Availability

Not applicable.

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
