# Peer review of "Cellular Toxicity Mechanisms and the Role of Autophagy in Pt(IV) Prodrug-Loaded Ultrasmall Iron Oxide Nanoparticles Used for Enhanced Drug Delivery"

_pharmaceutics, 2021, doi:10.3390/pharmaceutics13101730_

Round 1
Reviewer 1 Report
The introduction paragraphs are too big, if it possible try to make the introduction more straight to the goal of the paper.
Page 2, line 69. Please, rewrite the whole paragraph. What are all the possibilities that should be fulfilled by the paper?
Line 109. How they proved to be stable?
Please, add what type of the cell is A2780.
Line 117. Remove the bold from the phrase.
Line 148. Add from on “ranging 0 to 30”.
Line 155. What treatments were done?
Fig 1A. Is it possible to have a better quality picture? It is hard to see where is the cell borders. Please add on the methodology how the TEM was conducted. Does the authors have more pictures that can prove the conclusions?
Please, enhance the quality of all graphs.
Line 221. Please, add references that corroborate the findings or compare the data with similar experiments on literature.
Figure 2. Where is the insert b on the graph?
How the authors explain the increasing in cell viability on 10 µM at t=0h?
Author Response
Please, see the attachment

Reviewer 2 Report
This manuscript prepared the Pt-Fe NPs to enhance the cellular toxicity and autophagy against cancer. However, several issues should be addressed , which were listed as following:
- There have been a lot of references on Pt-loaded NPs. Compared with the reported Pt-loaded NPs , the advantages of Pt-Fe NPs should be discussed. And The following related references should be cited and discussed, such as
CHINESE CHEMICAL LETTERS 2019, 30 (1) , pp.243-246, CHINESE CHEMICAL LETTERS 2019 30 (11) , pp.1942-1946
- The release profiles of Pt for Pt-Fe NPs should be measured.
- To study the autophagy and apoptosis of cancer cells induced by Pt-Fe NPs, the blank Fe NPs at save concentration also should be set as control group.
Author Response
Please, see the attachment

Reviewer 3 Report
In this work Gutiérrez-Romero and co-authors described how ultrasmall iron nanoparticles loaded with cisplatin prodrug can enhance the delivery of the drug to cells and try to explain the toxicity mechanisms like authophagy. I found that the work largely overlapping with another published by the same authors (Doi: 10.1016/j.aca.2021.338356). To make it more innovative I suggest to insert new experiments especially regarding the autophagy section, or test the activity of nanoparticles on different types of cancer cells. Therefore I suggest accepting the job only after major revisions. Here you can find some comments:
-A full characterization of nanoparticles (TEM/SEM, DLS measurements) must be provided.
-This two works should be added to the references since a huge works were done by Voliani et al., on nanostructurs for delivery of cisplatin prodrug (IV) to tumors. Doi: 10.1002/ppsc.201600175 and 10.3390/cancers12051063.
-Is UL-trasmall in the title spelled correctly? I did not found the same spell in the text.
-What do you mean with “labelled nanoparticles”? please specify.
-Line 194, please specify what Fexp and Fcri state for.
-TEM image is not clear. Images with higher resolution should be provided. You should add a magnification of the cell in which is visible nanoparticles inside the cytosol surrounded by membrane after endocytotic internalization. Since internalization by early endosomes is indicated in the main text, this must be highlighted in the TEM image. Moreover, seems that the majority of nanoparticles remain outside of the cells, please clarify this point. Finally, the procedure to prepare TEM samples was not described in the materials and methods section.
- Figure 1B is not described at all in the main text. Please add the description.
-Figure 4A, there are no labels that specify which treatment was applied to cells.
-The effect on viability of iron nanoparticles after 24h treatment and 24h resting should be analyzed also for apoptosis/necrosis and cell cycle experiments to demonstrate that cells acquire resistance to the drug.
- Since a lot of nanoparticles seems to remain outside the cells how the author can be sure that the effect is not due to the prodrug release in the medium and subsequently internalized inside cells?
Author Response
Please, see the attachment

Round 2
Reviewer 3 Report
The authors repled adequately to all the points raised and therefore I suggest to accept the work